# Non-Destructive Evaluation of the Physiochemical Properties of Milk Drink Flavored with Date Syrup Utilizing VIS-NIR Spectroscopy and ANN Analysis

**DOI:** 10.3390/foods13040524

**Published:** 2024-02-08

**Authors:** Mahmoud G. Elamshity, Abdullah M. Alhamdan

**Affiliations:** Chair of Dates Industry & Technology, Agricultural Engineering Department, College of Food & Agricultural Sciences, King Saud University, Riyadh 11451, Saudi Arabia; melamshity@ksu.edu.sa

**Keywords:** milk, dates, syrup, Sukkary, Khlass, drink, modeling, VIS-NIR, ANN

## Abstract

A milk drink flavored with date syrup produced at a lab scale level was evaluated. The production process of date syrup involves a sequence of essential unit operations, commencing with the extraction, filtration, and concentration processes from two cultivars: Sukkary and Khlass. Date syrup was then mixed with cow’s and camel’s milk at four percentages to form a nutritious, natural, sweet, and energy-rich milk drink. The sensory, physical, and chemical characteristics of the milk drinks flavored with date syrup were examined. The objective of this work was to measure the physiochemical properties of date fruits and milk drinks flavored with date syrup, and then to evaluate the physical properties of milk drinks utilizing non-destructive visible–near-infrared spectra (VIS-NIR). The study assessed the characteristics of the milk drink enhanced with date syrup by employing VIS-NIR spectra and utilizing a partial least-square regression (PLSR) and artificial neural network (ANN) analysis. The VIS-NIR spectra proved to be highly effective in estimating the physiochemical attributes of the flavored milk drink. The ANN model outperformed the PLSR model in this context. RMSECV is considered a more reliable indicator of a model’s future predictive performance compared to RMSEC, and the R^2^ value ranged between 0.946 and 0.989. Consequently, non-destructive VIS-NIR technology demonstrates significant promise for accurately predicting and contributing to the entire production process of the product’s properties examined.

## 1. Introduction

The date palm, scientifically known as *Phoenix dactylifera* L., is considered one of the oldest fruit-bearing trees on the planet. Date palm trees are mainly cultivated in the Middle East, North Africa, Central and South America, Southern Europe, India, and Pakistan [1,2,3]. According to recent statistics, world date production reached 9.8 million tons in 2021 [4]. Saudi Arabia is one of the leading producers of dates, estimated to produce 1.6 million tons from 34 million palm trees [5].

Date fruits contain therapeutic bioactive agents and functional compounds that display antioxidant, antidiabetic, anticancer, antimutagenic, anti-inflammatory, and antianemic properties [6]. Due to their high and balanced nutritional profile, date fruits are a good candidate for enhancing human health and food security. These fruits contain high amounts of multivitamins, minerals, phytochemicals, dietary fiber, carbohydrates, and protein, demonstrating great nutritional value [7]. As a result, there is a growing interest in developing food products that use dates as a rich source of nutrients. A systematic review of 215 studies highlighted the potential nutritional and pharmaceutical properties of dates, as well as other parts of the palm tree [8]. In Kuwait, researchers [9] examined the biochemical and dietary composition of five varieties of dates. The analysis revealed that the energy content ranged from 3513 to 368.35 kcal/100 g. The potassium content was rich, ranging from 4450 to 7128 mg/100 g, while the calcium content ranged from 287 to 469 mg/100 g. The magnesium content was found to be between 130 to 294 mg/100 g, the sodium content ranged from 70 to 123 mg/100 g, the iron content ranged from 2.5 to 3.2 mg/100 g, and the manganese content ranged from 0.7 to 1.5 mg/100 g. The major antioxidants found were Gallic (ranging from 5.8 to 20 mg/100 g) and chlorogenic (ranging from nell to 0.712 mg/100 g), while the ascorbic acid content ranged from 0.63 to 0.88 mg/1 mg. Dates are a rich source of carbohydrates, fiber, minerals, proteins, phenolic compounds, sterols, triterpenoids, carotenoids, and vitamins [9]. The bioactive constituents present in dates, such as polyphenols, flavonoids, carotenoids, triterpenoids, and sterols, offer various health benefits, including lowering hypercholesterolemia and glucose levels, as well as anticancer, antioxidant, antimicrobial, anti-inflammatory, and bone-stimulating activities.

Global food security issues require the creation of value-added products from locally available agricultural products with high losses. Although dates are still consumed as fresh fruit, their market prices are low. The losses of dates in the market chain are estimated to be up to 26%. Most date-producing countries still use traditional packaging for raw dates. Value-added date products can be a significant economic resource for producing and processing countries. There is a need to manufacture added-value products from second- or third-grade dates. Possible processed products include syrup, vinegar, liquid sugar, yeast, medical alcohol, and fodder. Date syrup is a superior natural sweetener concentrate that can be added to other foods, such as dairy products, for both nutrition and sweetness.

Milk is a significant dairy product with a long history of production and consumption. Its production has greatly increased over the past year, reaching the highest levels in recent decades. In 2021, global milk production reached 0.9 billion tons, with the majority (0.8 billion tons) produced by cattle, followed by buffalo (0.12 billion tons) and goats (20.7 million tons) [4]. The rise in global milk production can be attributed to its high nutritional and health value, as well as the increased variety of milk products available and the growing world population.

In recent years, milk production in KSA has significantly increased, reaching over 2.9 million tons per year in 2021, which is equivalent to over seven million liters daily, or more than eighteen million bottles per day. This has contributed more than SAR 7 billion to the KSA economy [4]. In Saudi Arabia, the primary sources of milk production are cattle, camels, goats, and sheep, with cattle being the most prominent source, producing 2.6 million tons per year from 700,000 cows [10]. According to reference [4], camels produce 135.3 thousand tons, goats produce 96 thousand tons, and sheep produce 84.8 thousand tons. The consumption of camel milk has increased due to its high nutritional value and significant health benefits [4,11]. The country has achieved over 121% self-sufficiency in fresh dairy products in 2021, thanks to the utilization of advanced dairy production and processing systems [12]. The integrated chain model of dairy production, from raw materials to the final product, has been implemented by a few companies globally and has been adopted by many dairy companies in KSA [12]. The operational process of these enterprises includes obtaining cow breeds that guarantee high milk production and quality. The production of different high-value products, such as cheese, butter, and yogurt, is of great economic importance. The dairy sector produces a range of highly nutritious products that are in high demand among consumers. These include pasteurized and long-life milk, yogurt, labneh, cheese, butter, ice cream, and flavored milk, such as chocolate, strawberry, vanilla, caramel, and coffee, to enhance the taste of milk and other dairy products. Flavored milk drinks available in supermarkets are typically made from milk with artificial or natural fruit flavors. Additionally, these milk drinks often contain added processed sugars.

In KSA, there are several large dairy factories that are fully integrated, from providing cow farms to transportation and marketing channels to nearby countries. The dairy industry can be divided into two sections: dairies that rely on fresh raw milk produced from breeds of cows adapted to the local environment, which is the largest section, and the second section, which depends on imported milk powder that is reconstituted [13,14].

Dairy milk is an essential component of a balanced diet throughout the lifecycle, from childhood to adulthood. It contains high-quality protein and amino acids, as well as a wide range of essential nutrients, including minerals, such as calcium, iron, zinc, and phosphorus, and vitamins such as B complex, D, and A [15]. A moderate consumption of milk and dairy products can provide health benefits, such as protection against the development of cardiovascular diseases, cancers, and diabetes. This is due to the creation of bioactive peptides from milk by the gut microbiota, as well as the richness of milk in calcium and magnesium, which regulate glycaemia [16]. Several dairy products are produced and consumed globally in addition to milk. These products have been proven to have nutritional and health benefits, leading to an increase in their production in recent years. Examples of dairy products include cheese, yogurt, fermented milk drinks, flavored beverages, whey protein supplements, and protein concentrate and isolate. They are produced in large quantities and have various food, nutritional, and health promotion applications [17]. The consumption of dairy products has been linked to numerous health benefits, including the prevention of obesity, cancer, tooth decay, and diabetes, as well as the reduction of LDL cholesterol and blood pressure, and the promotion of bone and muscle growth [18].

The availability of raw milk and dates in Saudi Arabia suggests the potential to create a unique drink that is nutritious, healthy, and naturally sweetened. One such value-added product is a milk drink flavored and sweetened with date syrup. This would create new marketing opportunities for dates and dairy products, providing an exclusive nutritional and health drink for consumers, particularly children in schools [19]. The combination of fresh milk and dates can compete with the synthetic chocolate, vanilla, coffee, caramel, banana, or strawberry flavored milk drinks currently available.

Recently, near-infrared spectroscopy (NIR) has become a widely adopted non-destructive method for swiftly evaluating various food properties. Numerous studies have attempted to predict the physical and chemical attributes of food products, whether fresh or during processing and storage, using NIR [20,21,22,23,24,25,26,27]. Additionally, researchers have investigated the correlation between spectroscopy and sensory evaluations of food items [28,29,30,31,32]. The combination of NIR and statistical analysis is a powerful method for revealing food properties and sensory evaluations. Creating a quality index model for food characteristics and then using NIR to express this quality index can simplify and accelerate food quality assessments during production and shelf-life stages [33,34,35,36,37,38,39,40,41,42,43,44].

Near-infrared spectroscopy (NIR) is a non-destructive technique that can be used effectively to assess the quality characteristics of food properties [45]. Visible–near-infrared models ranging from 300 to 2000 nm were used to estimate the concentration of soluble sugars in apple fruits. The models showed a relatively good R^2^ (0.91 to 0.97) [46,47,48,49]. For cherry fruits, the soluble sugar content was assessed using NIR spectra at wavelengths of 600–1100 nm, with a standard error of prediction (SEP) of 0.75 [50]. Kiwi fruits were evaluated for physical parameters of soluble solids concentration (SSC) and hue angle using near-infrared estimation, with R^2^ values of 0.82 and 0.93, respectively [51]. NIR spectroscopy was used [52] to classify Shahani dates into four maturity phases: Kimiri, Khalal, Rutab, and Tamr, based on their moisture content and TSS levels. The predicted models had R^2^ values of 0.98 and 0.96, respectively. According to Gَomez [53], a mandarin has six broadband peaks on its absorption curve. Spectroscopic analysis performed in the near-infrared (NIR) region revealed a significant absorption peak at 672 nm, suggesting the presence of pigments such as chlorophyll, which gives the fruit its distinctive green color.

Feng [54] attributed the physical properties of foods to the electromagnetic energy that each product can absorb at specific wavelengths, such as red (475 nm), blue (650 nm), and green (510 nm). NIR was successfully used to correlate several quality indices, including acidity, ascorbic acid, soluble sugar content, dry matter, and firmness, for various vegetables and fruits [45,47,55,56].

Tian et al. [57] developed an NIR model to predict three flavonoid components (total flavonoids, phloridin, and trilobatin) in sweet tea leaves. The R^2^ values of the NIR models were 0.967 for calibration, 0.858 for validation, and 0.818 for prediction sets, indicating the rapid and convenient ability of NIR to determine flavonoid content in sweet tea. Zhang et al. [58] established correlations between green tea components, such as caffeine, epigallocatechin-3-gallate (EGCG), and moisture content, using NIR during production. This highlights the model’s usefulness for the online monitoring of tea product quality during production processes. The authors emphasized the advantages of NIR analysis, including its speed, non-destructiveness, and lower sample size requirements compared to traditional HPLC analysis.

Pandiselvam et al. [59] investigated various spectroscopic techniques, including NIR, to assess the oxidative stability, detect adulteration, and identify harmful additives, pathogens, and toxins in coconut products. These non-destructive methods, including NIR, were successful in authenticating and determining the quality of coconut products, including the identification of adulterants in coconut oil. Araújo et al. [60] found strong correlations between the NIR spectra of green coffee beans and experimental values of electrical conductivity and potassium leaching. They obtained impressive R^2^ values of 0.97 and 0.88 for electrical conductivity and potassium leaching, respectively. Cano-Reinoso et al. [61] used NIR spectroscopy to assess and establish a linkage between the chemical composition, specifically α-guaiene and azulene, of 84 samples of patchouli aromatic oil from Indonesia. The calibration model that proved most effective was based on the second derivative. It achieved an R^2^ value exceeding 0.90 and a coefficient of variation below 2.98%, confirming the applicability of NIR for assessing the quality of patchouli oil. In a noteworthy study, Ghooshkhaneh et al. [62] used NIR spectra to detect citrus black rot disease caused by fungi. During the first, second, and third week post-inoculation, oranges inoculated with Alternaria were assessed at wavelengths ranging from 200 to 1100 nm. The accuracy of distinguishing healthy and infected samples was 60% in the first and second weeks and reached 100% in the third week.

Although PLS (partial least squares) is a traditional linear tool in chemometrics, ANNs (artificial neural networks) provide a robust alternative for the modeling of complex nonlinear relationships between input and output data. ANNs represent a relatively recent introduction in this field and find diverse applications in chemometric analysis, including mapping, regression, modeling, clustering, and classification [63,64]. Their ability to interpret and quantify overlapping peaks and to reduce the effects of interferences in mixed spectra makes them particularly valuable in the investigation of food products [65,66].

To date, there have been no published reports that quantify both objective and subjective measurements of the properties of milk drinks flavored with date syrup using VIS-NIR and ANN techniques. The aim of this research was to develop and evaluate the physiochemical properties of milk drinks (cow and camel) flavored with date syrup (Khlass and Sukkary) at different percentages, using VIS-NIR spectra with PLSR and ANN analysis.

## 2. Materials, Methods and Production

### 2.1. Materials

Two of the most common date varieties (Sukkary and Khlass) produced in Saudi Arabia were obtained from a local date market in Riyadh and stored at a temperature of 5 °C until the start of the experiments.

Full-fat milk (both cow and camel) that was pasteurized by a brand-name dairy company was bought from a local market in Riyadh region. Milk samples were delivered in cold containers to the Food Engineering Laboratory (FEL) and kept refrigerated for a maximum of two days until the start of the experiments.

### 2.2. Methods

#### 2.2.1. Date Syrup Production

Date syrup was produced utilizing a lab-scale level at FEL. The major unit operations include the removal of date pits, mechanical mincing to produce date paste, and mixing and heating of the date paste with water (at a *w*/*w* ratio of 2.5:1 (paste: water (weight/weight)) to produce a homogenized date suspension [67,68]. This was followed by press filtration of the date suspension to produce clarified date juice (Seitz Lab Filter Press, Model A20z, K150, Manufactured by Pall Fitter systems GmbH, BadK, Reuznach, for Pall International Sarl, Fribourg, Switzerland, Germany). Date juice was then concentrated under vacuum using a lab-scale rising film natural convection evaporator (QVF Teaching system, CTSY Evaporation, Climbing Film and Natural Circulation Evaporator, QVF, Switzerland, Germany) to produce high-quality date syrup. Date syrup can be preserved for several months at room temperature since sugar content reaches 80% [68,69,70,71,72,73,74,75].

#### 2.2.2. Preparation of Milk Drinks Flavored with Date Syrup

At the start of experiments, sixteen batches of milk flavored with date syrup were prepared as shown in Figure 1. Four amounts of date syrup (5, 10, 15, and 20 g) from two date cultivars (Sukkary and Khlass) were added to milk (95, 90, 85, and 80 g, respectively for cow and camel) so that the total milk drink batch would be 100 g. The initial sensory evaluation revealed that four batches were favorable from the sixteen mixes as reported in detail by Elamshity and Alhamdan [68,75]. The chemical and physical measurements were then performed on those four favorable mixtures which were 15% milk drink flavored with Sukkary date syrup, and 10% milk drink flavored with Khlass date syrup for both cow and camel milk.

The drink samples were then pasteurized in a water bath (Kenwood, KM070 Series, and the UK) at 75 °C for 5 min [68,76]. The samples were then cooled in an ice-water bath with a mixer (Cofrimell & Coldream 2 M, Via delle Monachelle, 66/b 00071 Pomezia—Rome, Italy) to 5 °C. It was then ready to conduct physiochemical experiments as well as sensory assessment.

#### 2.2.3. Physiochemical Evaluation

(a)Physical properties

The standard method for measuring the moisture content was conducted for the four date syrup-flavored milk drinks [77]. A vacuum oven (Vacutherm model VT 6025, Heraeus Instrument, D-63450. Hannover, Germany) was used to dry samples at a temperature of 70 °C under a vacuum of 200 mmHg for 48 h. Aqua Lab instrument (Model Series 3, Decagon Devices, Inc. Pullman, Washington, DC, USA) was used to measure water activity at room temperature (25 °C). The total soluble solids (TSS) of samples was measured using a refractometer (ABBE 5, Bellingham & Stanley Ltd., Longfield Rd, Tunbridge Wells, UK). The pH values of the samples were measured using a pH meter (Jenway, model 3510, Bibby Scientific Ltd., Stone, Staffordshire, UK) with measuring accuracy ± 0.01. After each measurement, the electrode was standardized using a pH 7.0 buffer. The total soluble solids and refraction indices of the samples was estimated using a refractometer (ABBA 5 (code 44-501), Bellingham and Stanley Ltd. (BS), Jena, Germany) at lab temperature (25 °C). The experiment comprised 120 replicates for each property, and all replications used the mean value calculation.

The densities of the sample drinks were determined at different temperatures (from 5 to 80 °C) using a density meter (DMA4100M, Anton Paar GmbH, Graz, Austria).

The basic color coefficients (L*, a*, and b*) of the samples were estimated using a color instrument (Color 45/0, Hunter Associates Laboratory, Inc., Thomas Mittelstadt, Sales Manager, 11491 Sunset Hills Road, Reston, Virginia, VA, USA). L* indicates whiteness or brightness/darkness, a* indicates redness/greenness, b* indicates yellowness/blueness, and L_0_*, a_0_*, and b_0_* are the corresponding values at zero time. The experiment comprised 120 replicates for each property, and all replications used the mean value calculation. Color derivatives of Chroma, Hue Angle, and Browning Index (BI) were calculated from the following basic color coefficients [78,79,80]:(1)$$Chroma={\left(a^{*2}+b^{*2}\right)}^{0.5}$$
(2)$$Hueangle=\tan^{-1}\left(\frac{b^{*}}{a^{*}}\right)$$
(3)$$BI=\frac{\left[100\left(x-0.31\right)\right]}{0.17}$$
where
$$x=\frac{\left(a^{*}+1.75L^{*}\right)}{\left(5.645L^{*}+a^{*}-3.012b^{*}\right)}$$

(b)Chemical analysis

Chemical analysis was performed for the fresh milk (cow and camel), date syrup (Khlass and Sukkary), and the four date syrup-flavored milk drinks. The chemical analysis was conducted according to the standards of the Association of Official Analytical Chemists [77]. The experiment comprised 10 replicates for each property, and all replications used the mean value calculation.

(c)Sensory evaluation

Sensory evaluation was carried out in two steps. The first was to evaluate the best drinks formulated from date syrup (5 to 20%) of the two date cultivars (Sukkary and Khlass) and add cow’s milk and camel’s milk as described and detailed in [68,75]. A nine-point structured-hedonic scale [81,82,83,84,85,86,87] was used, with a score of 9 indicating “like extremely”, and 1 meaning “dislike extremely”. The sensory characteristics were taste, flavor, texture, color, and general acceptance of the product. Thirty judges (professors, employees, and students from the departments of the Agricultural Engineering and Food Science and Nutrition, Food Science and Agriculture College, King Saud University) scored the sensory characteristics.

#### 2.2.4. Evaluation of Physical Properties of Milk Drinks Utilizing VIS-NIR

The non-destructive assessment of samples properties was conducted with visible–near infrared spectroscopy (VIS-NIR) using a handheld NIR meter (F-750, Firmware v.1.2.0 build 7041, Felix Instruments, Camas, WA, USA).

The F-750 system was equipped with a Zeiss MMS1 VIS-NIR spectrometer operating in the range of 310–1100 nm at 3 nm intervals. Prior to sample physical properties measurements, optical readings were taken for all samples. The F-750 device featured a reference shutter, allowing for the calculation of dark current and ambient light effects with each measurement, particularly when scanning with the lamp off. For each sample, three hundred scans were recorded and averaged. During the validation phase, spectra were acquired for a distinct set of samples under the same conditions at 5 °C.

VIS-NIR spectra tend to have linear baseline increases and these are removed by second derivatives which have negative peaks where the original had a peak, and are thus more readily comprehensible. For these reasons second derivatives are often preferred. The second derivative gives us a mathematical way to tell how the graph of a function is curved. The second derivative tells us if the original function is concave up or down.

Following spectrum capture, the data were transferred from the F-750 SD-Card to a PC for analysis. The stored data were imported and saved as a comma-separated values file (CSV file) using Data Viewer Software, Version 2.1.1. Spectra underwent pre-processing via the Savitzky–Golay second derivative. Calibration models were then established using a dataset comprising 3600 samples, 2520 spectra (training), 720 spectra (test), and 360 spectra (validate). Two analysis tools were employed for calibration model development: partial least-squares regression (PLSR) and artificial neural network (ANN).

PLSR entailed examining the spectra as a linear multivariate relationship [88]. This method involved identifying latent variables (LVs) that elucidate the most significant variance in the data. LVs were discerned by exploring subspaces enhancing covariance between predictor and response variables [89].

In parallel, artificial neural networks (ANNs) were employed, providing a robust nonlinear pattern recognition approach capable of modeling intricate diversities, environmental factors, and instrument variations [90]. The estimated data were analyzed using App-Builder v2.1.7 software (Felix Instruments, Camas, WA, USA) [91,92]. Evaluation of prediction performance relied on calibration and validation results, assessed with correlation coefficient (R^2^), root-mean-square error in calibration (RMSEC), and cross-validation (RMSECV) [22]. 

### 2.3. Statistical Analysis

All quantifiable properties were subjected to analysis using a statistical software package (SAS software, Version 9.2, SAS Institute Inc., Cary, NC, USA) [93]. Experimental data were utilized for the analysis of variance (ANOVA). The data presented in the manuscript’s tables represent the averages derived from fifty replicates obtained through physicochemical measurements. The least significant difference (LSD) was calculated at *p* ≤ 0.05. The prediction performance was assessed based on calibration and validation results utilizing AppBuilder v2.1.7 software (Felix Instruments, Camas, WA, USA). Graphs, plots, and other calculations were performed using the Microsoft Office 365 package (Microsoft, Redmond, WA, USA).

## 3. Results

The approach of this study was to produce date syrup and then add it to formulate milk drinks flavored with the syrup. The milk drinks would be evaluated for two cultivars of syrup, two types of milk, and at different added concentrations. From the objective and subjective assessments of the selected milk drinks, VIS-NIR models would be developed.

All data of the physicochemical properties and sensory evaluation of the date syrup, fresh milk, and the four preferred milk drinks are presented in the following sections.

### 3.1. Physical Properties (MC, aw, TSS, Color, Density, pH)

Date syrup was successfully produced at the lab-scale level. The basic physical properties (moisture content, water activity, total soluble solids, and pH) were experimentally measured for the date syrup, fresh milk, and preferred milk drinks flavored with date syrup, and are shown in Table 1. Although the values appeared to be similar within each group, the variations were significantly different within date syrup, fresh milk, and milk drinks, except for the TSS of date syrup. There were variations in the properties of the four date syrup-flavored milk drinks. The initial values of fresh milk and date syrup had an apparent effect on the milk drinks’ properties. It can be noticed that fresh milk had a notable effect on milk drinks properties due to the high percentage of milk (85 to 90%) compared to that of syrup (10 to 15%). The experimental results for the four drinks are summarized in Table 1 for mean values and variance analysis.

#### 3.1.1. Color

Table 2 shows the mean values of the basic color parameters (L*, a*, b*) and their derivatives parameters (Chroma, Hue Angle, and BI). They were measured at room temperature (25 °C) for the date syrup, fresh milk, and preferred milk drinks.

It can be seen from Table 2 that there were variations and significant differences within date syrup, fresh milk, and milk drinks due to the nature of milk and dates, in addition to the storage and processing parameters. It is apparent that the (L*) values were higher for fresh milk (whiteness) compared to syrup (darkness). This shows the importance of the (L*) parameter compared to the (a*) and (b*) parameters for the samples examined in this study.

The mean values of the basic parameter (L*) of the milk drinks, which ranged between zero for darkness and one hundred for whiteness (brightness) varied from 85.133 to 88.667. This indicated the effect of the date syrup (dark brown) on darkening the color of the milk drinks. The mean value of (a*) varied from −0.075 to 0.418, whereas the mean value of (b*) varied from 13.383 to 15.262.

#### 3.1.2. Density

Densities for date syrup, fresh milk, milk drinks samples were measured at a temperature range from 5 to 80 °C, as shown in Table 3. It is interesting to note that all samples’ densities were higher than 1.00, except that of the camel milk at 80 °C. The density values for all samples decreased with the increase of temperature.

The mean values of density values for the four milk drinks within a temperature range of 5 to 80 °C are shown in Figure 2. It is apparent that the density values for all four samples decreased with increasing temperature. Over the experimental temperature range, the average density values ranged from 1.030 to 1.079 g/cm^3^ (Table 3).

The densities of each sample of fresh milk, syrup, and milk drink were well-correlated linearly with temperature (R^2^: 0.970 to 0.988), as shown in Table 4.

### 3.2. Sensory Evaluation of All Samples of Milk Drinks Flavored with Date Syrup

The sensory evaluations of the milk drinks were conducted by sixty participants, using a nine-point structured hedonic scale for taste, flavor, texture, color, and overall acceptability. The tests were conducted for cow’s and camel’s milk flavored with date syrup (Sukkary and Khlass cultivars) at different concentrations. A summary of the sensory evaluation results for the four milk drinks is illustrated in Figure 3. The results indicated that the cow’s milk flavored with 10% Khlass syrup was significantly (α = 0.05) preferred as the favorable milk drink.

### 3.3. Chemical Analysis of Fruits, Milk, Syrup, and Milk Drinks

The analysis data for the chemical composition of the fruit flesh of dates (Sukkary and Khlass) are shown in Table 5. Probably the most noticeable observation in comparing dates of the two cultivars is the phosphorus content, with that of Sukkary dates being double that of Khlass dates. The second notable result was the percentage of sugars in Sukkary dates (7.8, 8.9, 63.4 g/100 gm dm) compared to Khlass dates (42.3, 54.5, 0.3) for glucose, fructose, and sucrose sugars, respectively.

Table 6 shows the chemical composition of fresh milk, date syrup, and the four milk drinks. There were significant differences between the average experimental values of the chemical components of the four date syrup-flavored milk drinks. Vitamin (A) was in the range of 37.13 to 556.5 IU/100 g and 10.00 IU/100 g for vitamin D. The total energy varied between 78.50 and 99.00 kcal/100 g. The concentrations of the most important metals ranged between 867 and 1002 ppm for calcium, 106 and 142 ppm for magnesium, 289 and 372 ppm for sodium, and 1702 and 2242 ppm for potassium.

### 3.4. Visible–Near Infrared Spectroscopy (VIS-NIR)

Mean second absorbance derivative spectra were generated for the fresh milk, date syrup, and milk drinks, consisting of 1200 samples for each of them within the wavelength range of 729–975 nm to examine the physical properties of M.C., aw, TSS, pH, and BI. Two analytical tools were utilized: partial least-squares regression (PLSR) and artificial neural networks (ANNs).

The PLSR performance in calibration and cross-validation for the M.C., aw, TSS, pH, and BI properties for milk drinks is shown in Table 7. In the calibration models, the R^2^ and RMSEC values were 0.988 and 0.777 for M.C., respectively; 0.984 and 0.746 for aw, respectively; 0.941 and 0.727 for TSS, respectively; 0.957 and 0.755 for pH, respectively; and 0.978 and 0.743 for BI, respectively.

The performance of ANNs in calibrating and cross-validating for the properties of the favorable drink (cow’s milk with 10% Khlass) is summarized in Table 8. In the calibration models, the R^2^ and RMSEC values were 0.989 and 0.745 for M.C., respectively; 0.984 and 0.755 for aw, respectively; 0.946 and 0.715 for TSS, respectively; 0.955 and 0.740 for pH, respectively; and 0.978 and 0.735 for BI, respectively. In the cross-validation assessments, the R^2^ and RMSECV values were 0.989 and 0.744 for M.C_D_, respectively; 0.984 and 0.725 for aw, respectively; 0.946 and 0.754 for TSS, respectively; 0.955 and 0.711 for pH, respectively; and 0.978 and 0.713 for BI, respectively.

Figure 4 shows the spectra absorbance for the fresh milk, Khlass date syrup, and the favorable milk drink (cow’s milk with 10% Khlass) utilizing the Flex F750 instrument.

The second derivative of NIR spectra was introduced to enhance data quality. Specifically, the mean second absorbance derivative spectra were generated for the fresh milk, Khlass date syrup, and the favorable milk drink (cow’s milk flavored with 10% Khlass) consisted of 3600 samples within the wavelength range of 390–1029 nm, as illustrated in Figure 5. It can be noted that the spectra behavior of both fresh milk and milk drinks were closer compared to date syrup apparently due to the low amount of syrup in the milk drink. In the figure, the curves for each color represent 1200 samples.

## 4. Discussion

### 4.1. Physical Properties (MC, aw, TSS, Color, Density, pH)

It can be observed that the moisture content of Sukkary date syrup was higher than that of Khlass. This might be due to the initial moisture content of the Sukkary dates (9.5%wb), while it was 6.7%wb for Khlass. A similar trend was observed for water activity. The average moisture content showed significant differences, with the camel’s milk flavored with Khlass date syrup (10%) being higher than that of the other three samples.

The water activity and pH of cow’s milk flavored with Khlass date syrup (10%) were higher than those of the other three samples. The TSS was found to be higher in cow’s milk flavored with Sukkary date syrup at 15%, which can be attributed to a lower moisture content and higher amounts of date syrup.

From basic color parameters, derivative parameters can be obtained. The mean values of the derivative color parameters (Chroma, Hue angle, and BI) for the samples ranged from 13.383 to 15.267, −57.426 to 57.474, and 15.915 to 19.630, respectively. These data showed significant differences in the derivative color parameters of milk drinks. The differences might be attributed to the initial color of the raw date fruit milk. The BI values for the four milk drinks were higher than that of the raw milk but lower than that of date syrup (as shown in Table 2). The decrease in BI values of the four milk drinks was due to the high amount of milk present.

The density of all samples decreased with the increase of temperature due to the expansion of the sample volume. The elevated temperature leads to the weakening of the forces between molecules bonds, and thus the molecules of the sample expanded. Thus, the effect of temperature on density and volume should be considered in the design of pasteurization and sterilization processes of milk drinks flavored with date syrup.

### 4.2. Sensory Evaluation of All Samples of Milk Drinks Flavored with Date Syrup

The results of the sensory evaluation revealed that the most favorable milk drink was cow milk flavored with Khlass date syrup (10%) followed by cow milk with Sukkary date syrup (15%). The third preference was camel milk with Sukkary date syrup (15%) and lastly camel milk with 10% of Khlass date syrup. Detailed results of the sensory evaluation can be found in [68,75]. They showed that the most flavored milk drink was chosen from sixteen milk drinks (mix of 5 to 20%). The favorable drink was cow’s milk flavored with 10% Khlass, as indicated by its taste, flavor, texture, color, and overall acceptability (7.200, 7.333, 7.467, 7.267, and 7.200, respectively) The results were consistent with a number of previous research [68,75]. There is a need for further sensory investigation to include students from different grades, and additional sectors of the society (probably based on age, gender, location, and education). Furthermore, it is recommended to evaluate such milk drinks with sensory evaluation along the shelf-life of the pastoralized and sterilized milk drinks.

### 4.3. Chemical Analysis of the Milk Drink with Date Syrup

The most pronounced differences of the milk drinks were in the total carbohydrates, fructose, glucose, sucrose, maltose, casein, fat, ash, and total energy. These differences were due to the chemical components of both the date cultivar and milk type. Milk with Sukkary date syrup at a relatively higher concentration (15%) increased the total carbohydrate, sucrose, maltose, and total energy compared with that of milk with Khlass date syrup at 10%. The mean values of the chemical components in the four milk drinks varied in the range of 12.61–16.09 g/100 g for total carbohydrates, 2.60–3.95 g/100 g for fructose, 1.84–4.18 g/100 g for glucose, 0.10–5.91 g/100 g for sucrose, 0.45–1.41 g/100 g for maltose, 3.40–4.02 g/100 g for lactose, 2.88–2.93 g/100 g for proteins, 1.87–2.30 g/100 g for Casein, 1.75–2.78 g/100 g for fat, 0.60–0.86 g/100 g for ash, and 0.21–0.26 g/100 g for standard acidity (lactic acid). These variations in the milk drinks’ composition and sensory evaluation must be considered in the marketing, nutritional values, and total cost for each selected milk drink. The results were consistent with a number of previous studies [68,75].

### 4.4. Evaluating Physical Properties Utilizing VIS-NIR Technique

Recently, near-infrared spectroscopy (NIR) has gained more attention with the advancement of high-speed computers. It has wider applications in identifying food properties. One valuable technique for eliminating extraneous signals from the spectra involves the application of derivatives of NIR spectroscopy [94,95].

Several methods of pre-processing can be implemented prior to modeling [22,96]. After the completion of NIR experiments, two analytical tools can be utilized: PLSR and ANNs.

In this study, PLSR revealed that the R^2^ and RMSECV values were 0.982 and 0.788 for M.C.; 0.996 and 0.764 for aw; 0.687 and 0.727 for TSS; 0.955 and 0.723 for pH; 0.988 and 0.703 for BI, respectively. For ANNs, the correlation coefficient (R^2^) values fell within the range of 0.942 to 0.996, indicating a particularly good model performance. This range signifies an excellent model quality, where R^2^ values exceeding 0.70 are generally deemed satisfactory in NIR modeling [97,98,99]. These findings underscore the effectiveness of the PLSR and ANNs techniques as robust statistical tools for accurately predicting both the objective properties and subjective evaluation of foods, in this case milk drinks.

These results affirm the excellence of the ANNs technique in predicting the properties of milk drinks. It is evident that the correlation coefficients in the ANNs model surpassed those of the PLSR analysis. The RMSEC serves as a measure indicating how well the calibration model aligns with the calibration set, typically decreasing as the number of factors increases. However, the RMSECV, which represents the root-mean-square error in cross-validation, may occasionally rise with the addition of more factors. The RMSECV is considered a more reliable indicator of a model’s future predictive performance compared to RMSEC [100].

This study’s model for a favorable milk drink used the RMSECV compared to those for previously published studies, for example, Tange et al. [101], which only used RMSE for some food products such as juice, syrup, Massecuite, and molasses, which reached 0.16, 0.25, 0.47, and 0.79, respectively.

Thus, both analytical techniques, PLSR and ANNs, have demonstrated their effectiveness in modeling NIR data during the calibration and cross-validation phases in this study. Furthermore, the ANNs model was preferred for the present data over PLSR due to the higher correlation coefficients. Therefore, these analyses validate the strong alignment between VIS-NIR and the properties of the preferred milk drink.

In practical applications, a non-destructive handheld VIS-NIR spectrophotometry meter can be successfully employed by food producers, processors, and regulatory authorities throughout the entire production, processing, transportation, storage, and retail chain to assess a product’s quality. This technology holds the potential to serve as a foundational framework for further research and commercial applications. This utilization can be instrumental in monitoring product ‘quality’ and ‘shelf-life’, aligning with the standards set forth by relevant authorities, such as SFDA.FD 150-1/2018 [102].

## 5. Conclusions

Date syrup was produced successfully at the lab-scale level and can be utilized as a sweetener and has high nutritional value. The obtained date syrup (Sukkary and Khlass cultivars) was then added to cow’s and camel’s milk at different ratios (5, 10, 15, and 20%) constituting a combination of sixteen drinks (1200 samples) of milk flavored with date syrup. Of the sixteen drinks, four mixes were selected based on a sensory evaluation.

The study initially delved into the analysis of the physical and chemical composition of the four preferred milk drinks, which were all flavored with date syrup. These milk drinks exhibited noteworthy disparities in their chemical constituents, encompassing total carbohydrates, fructose, glucose, sucrose, maltose, casein, fat, ash, and total energy. The sensory evaluation discerned that the milk drink, combining cow’s milk with 10% Khlass date syrup, outperformed the other variants.

The resulting date syrup-flavored milk drinks represent a natural and highly nutritious sweet product, highlighting the potential to establish a new market that leverages locally sourced date and milk products. It is an aspiration that the findings of this study will foster the production of distinctive, natural, and nutritionally rich products on an industrial scale.

The application of near-infrared spectroscopy (NIR) facilitated the correlation between reflectance/absorbance and the physiochemical attributes of food products. Notably, NIR enables the direct measurement and tracking of both sensory and objective assessments through NIR analysis, employing both partial least-square regression (PLSR) and artificial neural networks (ANNs) approaches.

VIS-NIR proved effective in accurately predicting physiochemical properties and aligning well with measured sample attributes (moisture content, water activity, total soluble solids, pH, Browning index properties). The coefficient of determination (R^2^) ranged from 0.987 to 0.996, underlining the competence of VIS-NIR technology in estimating the physiochemical properties of the favorable milk drink with date syrup. These predictions were affirmed in both the calibration and cross-validation phases.

Consequently, non-destructive VIS-NIR spectrophotometry stands as a robust tool for forecasting physiochemical properties and holds significant potential for practical implementation across the entire food production, processing, transportation, storage, and retail chain.

## Figures and Tables

**Figure 1 foods-13-00524-f001:**
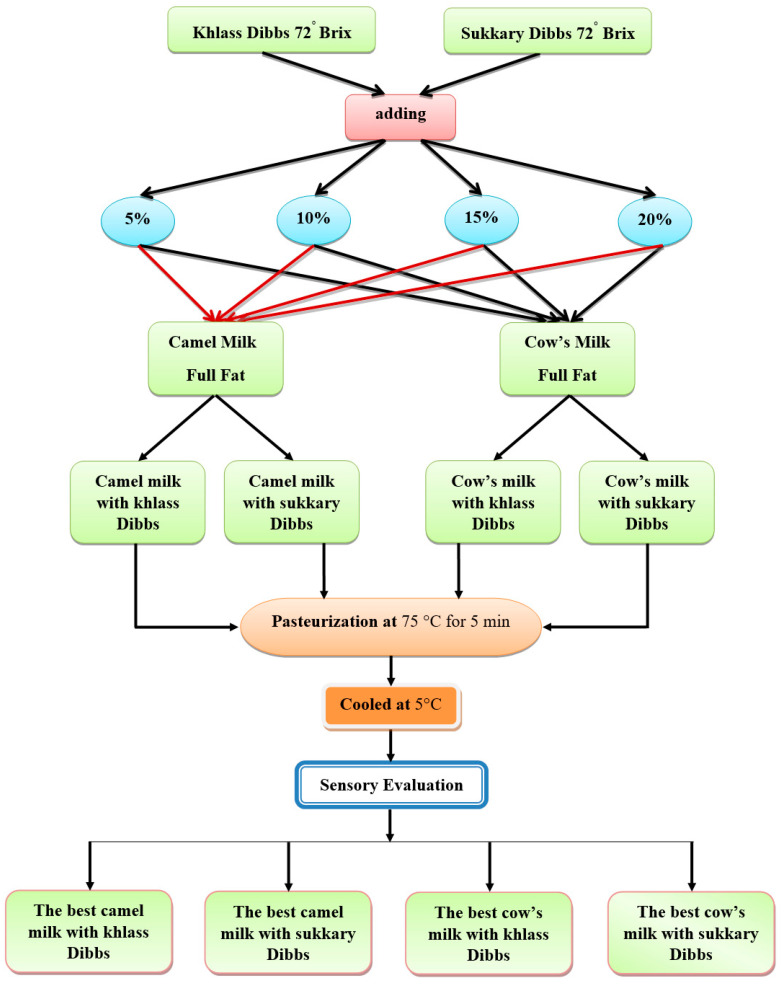
Preparation of milk and Dibb’s drinks.

**Figure 2 foods-13-00524-f002:**
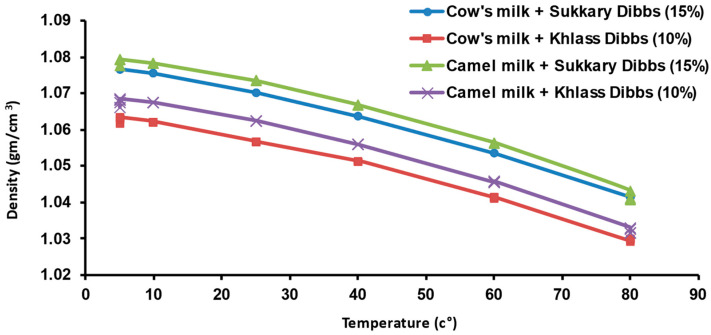
Density values for the four milk drinks at different temperatures.

**Figure 3 foods-13-00524-f003:**
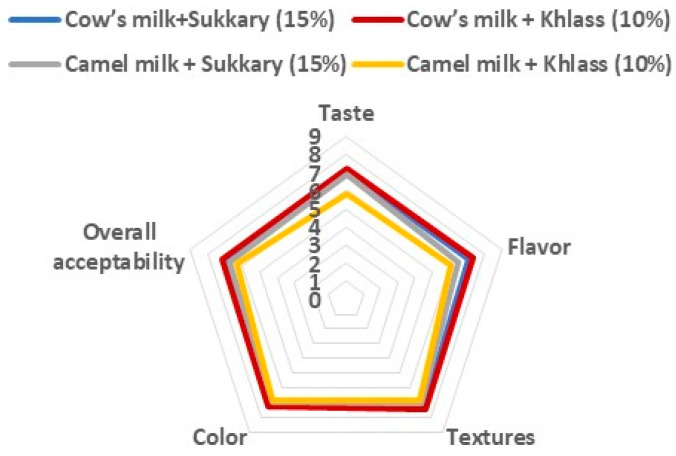
Sensory preference of the taste, flavor, texture, color, and overall acceptability for the four milk drinks.

**Figure 4 foods-13-00524-f004:**
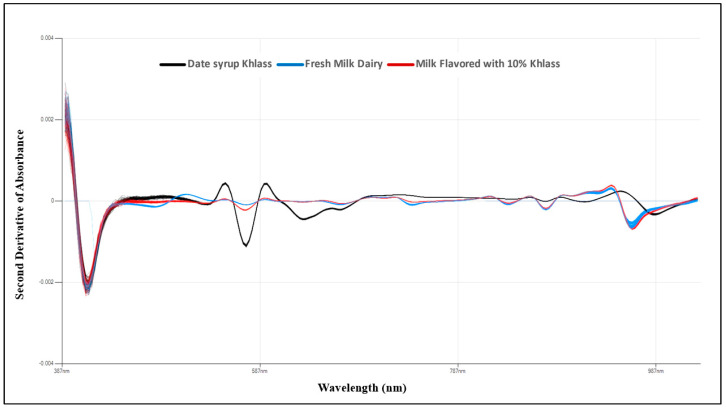
Image of raw spectra absorbance view of total of 3600 samples for the fresh milk, Khlass date syrup, and the favorable milk drink (cow’s milk with 10% Khlass), utilizing the Flex F750 instrument and Data Viewer Software.

**Figure 5 foods-13-00524-f005:**
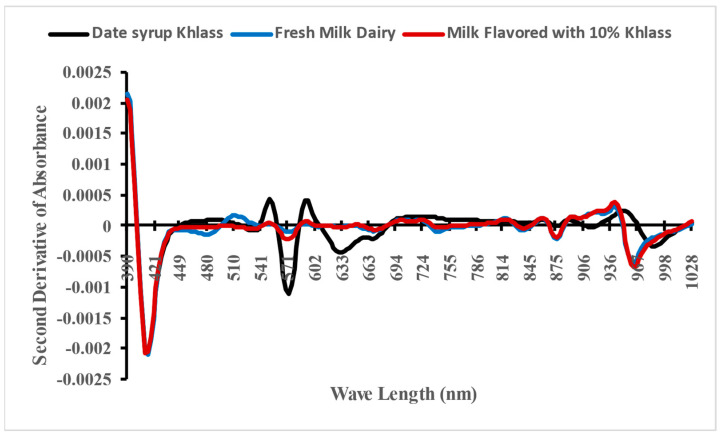
The average second derivative of absorbance within the wavelength range of (390–1029 nm) for the fresh milk, Khlass date syrup, and the favorable milk drink (cow’s milk with 10% Khlass).

**Table 1 foods-13-00524-t001:** Mean values of the basic physical properties of the date syrup, fresh milk, and the four preferred milk drinks.

Product Type	Moisture Content (w.b.) (%)	Water Activity	Total Soluble Solids (Brix°)	pH
Date syrup:				
Sukkary	26.263 ^a^ ± 0.652	0.765 ^a^ ± 0.005	72.200 ^a^ ± 0.001	4.944 ^b^ ± 0.081
Khlass	25.405 ^b^ ± 0.585	0.703 ^b^ ± 0.002	72.600 ^a^ ± 0.002	5.120 ^a^ ± 0.131
Milk:				
Cow	87.392 ^b^ ± 0.331	0.996 ^a^ ± 0.006	11.600 ^a^ ± 0.012	6.764 ^a^ ± 0.162
Camel	88.933 ^a^ ± 0.242	0.989 ^b^ ± 0.007	9.100 ^b^ ± 0.014	6.554 ^b^ ± 0.190
Favored milk drinks:				
Cow milk + Sukkary (15%)	77.134 ^d^ ± 0.173	0.977 ^b^ ± 0.004	22.86 ^a^ ± 0.003	6.533 ^b^ ± 0.009
Cow milk + Khlass (10%)	80.215 ^b^ ± 0.265	0.982 ^a^ ± 0.002	19.78 ^c^ ± 0.001	6.663 ^a^ ± 0.052
Camel milk + Sukkary (15%)	77.810 ^c^ ± 0.054	0.963 ^c^ ± 0.003	22.19 ^b^ ± 0.003	6.353 ^c^ ± 0.005
Camel milk + Khlass (10%)	80.831 ^a^ ± 0.222	0.977 ^b^ ± 0.010	19.16 ^d^ ± 0.002	6.535 ^b^ ± 0.008

The same letter in each column with each group indicates that the average values are not significantly different at *p* < 0.05.

**Table 2 foods-13-00524-t002:** Mean values of the basic and derivatives color parameters for the date syrup, fresh milk, and four preferred milk/date syrup drinks.

Product Type	Basic Color Parameters			Derivative Color Parameters		
	L*	a*	b*	Chroma	Hue Angle	BI
Date syrup Sukkary	4.444 ^b^ ± 0.026	1.673 ^b^ ± 0.024	5.348 ^b^ ± 0.016	5.604 ^b^ ± 0.017	51.730 ^b^ ± 0.093	339.689 ^a^ ± 0.495
Date syrup Khlass	6.892 ^a^ ± 0.046	2.231 ^a^ ± 0.114	7.588 ^a^ ± 0.075	7.910 ^a^ ± 0.069	52.103 ^a^ ± 0.323	277.498 ^b^ ± 0.298
Cow Milk	95.795 ^a^ ± 0.024	−2.113 ^a^ ± 0.023	9.064 ^a^ ± 0.035	9.307 ^a^ ± 0.029	−53.303 ^a^ ± 0.066	8.064 ^a^ ± 0.054
Camel Milk	94.912 ^b^ ± 0.040	−1.274 ^b^ ± 0.007	5.627 ^b^ ± 0.046	5.770 ^b^ ± 0.046	−53.432 ^b^ ± 0.038	4.978 ^b^ ± 0.052
Cow milk + Sukkary (15%)	86.128 ^c^ ± 0.267	0.038 ^c^ ± 0.02	13.991 ^c^ ± 0.244	13.991 ^c^ ± 0.244	57.474 ^a^ ± 0.024	17.346 ^c^ ± 0.402
Cow milk + Khlass (10%)	88.667 ^a^ ± 0.123	−0.075 ^d^ ± 0.006	13.383 ^d^ ± 0.217	13.383 ^d^ ± 0.217	−57.426 ^d^ ± 0.008	15.915 ^d^ ± 0.304
Camel milk + Sukkary (15%)	85.133 ^d^ ± 0.024	0.379 ^b^ ± 0.007	15.262 ^a^ ± 0.012	15.267 ^a^ ± 0.012	57.104 ^b^ ± 0.007	19.630 ^a^ ± 0.028
Camel milk + Khlass (10%)	87.873 ^b^ ± 0.024	0.418 ^a^ ± 0.006	14.444 ^b^ ± 0.058	14.450 ^b^ ± 0.058	57.034 ^c^ ± 0.006	17.895 ^b^ ± 0.086

The same letter in each column indicates that the average values are not significantly different at *p* < 0.05.

**Table 3 foods-13-00524-t003:** Mean values of density (g/cm^3^) of the date syrup (for the two cultivars) at different temperatures (between 5 and 80 °C), fresh milk (for the two types), and four preferred milk/date syrup drinks *.

Temperature, °C	5	10	25	40	60	80
Date syrup Sukkary	1.373 ^b^ ± 1.342 × 10^−4^	1.370 ^b^ ± 5.477 × 10^−5^	1.364 ^b^ ± 4.472 × 10^−5^	1.356 ^a^ ± 0.000	1.346 ^a^ ± 5.477 × 10^−5^	1.332 ^a^ ± 4.472 × 10^−5^
Date syrup Khlass	1.378 ^a^ ± 4.472 × 10^−5^	1.376 ^a^ ± 5.477 × 10^−5^	1.367 ^a^ ± 4.472 × 10^−5^	1.359 ^a^ ± 0.000	1.345 ^a^ ± 0.000	1.330 ^a^ ± 1.095 × 10^−4^
Cow Milk	1.034 ^a^ ± 4.472 × 10^−5^	1.033 ^a^ ± 4.472 × 10^−5^	1.029 ^a^ ± 0.000	1.023 ^a^ ± 4.472 × 10^−5^	1.013 ^a^ ± 4.472 × 10^−5^	1.010 ^a^ ± 5.477 × 10^−5^
Camel Milk	1.033 ^a^ ± 0.000	1.032 ^a^ ± 0.000	1.028 ^a^ ± 5.477 × 10^−5^	1.022 ^a^ ± 0.000	1.012 ^a^ ± 4.930 × 10^−4^	1.000 ^a^ ± 2.168 × 10^−4^
Cow milk + Sukkary (15%)	1.077 ^b^ ± 4.472 × 10^−5^	1.076 ^b^ ± 5.477 × 10^−5^	1.070 ^b^ ± 0.000	1.064 ^b^ ± 0.000	1.054 ^b^ ± 0.000	1.041 ^a^ ± 0.000
Cow milk + Khlass (10%)	1.063 ^d^ ± 5.958 × 10^−4^	1.062 ^d^ ± 5.477 × 10^−5^	1.057 ^d^ ± 4.472 × 10^−5^	1.051 ^d^ ± 2.510 × 10^−4^	1.041 ^d^ ± 4.472 × 10^−5^	1.031 ^c^ ± 2.683 × 10^−4^
Camel milk + Sukkary (15%)	1.079 ^a^ ± 7.950 × 10^−4^	1.078 ^a^ ± 0.000	1.074 ^a^ ± 0.000	1.067 ^a^ ± 4.472 × 10^−5^	1.057 ^a^ ± 0.000	1.041 ^a^ ± 1.026 × 10^−3^
Camel milk + Khlass (10%)	1.068 ^c^ ± 9.094 × 10^−4^	1.068 ^c^ ± 0.000	1.063 ^c^ ± 0.000	1.056 ^c^ ± 0.000	1.046 ^c^ ± 1.304 × 10^−4^	1.032 ^c^ ± 7.301 × 10^−4^

* The same letter in each column within a group indicate that the average values are not significantly different at *p* < 0.05.

**Table 4 foods-13-00524-t004:** Linear density prediction equations and their constants and correlation coefficients for the date syrup, fresh milk, and four milk drinks valid for a temperature range of 5 to 80 °C.

Product Type	ρ = a T + b		R^2^
	a	b	
Date syrup Sukkary	−0.0005	1.376	0.992
Date syrup Khlass	−0.0006	1.383	0.995
Cow Milk	−0.0004	1.038	0.978
Camel Milk	−0.0004	1.037	0.972
Cow milk + Sukkary (15%)	−0.0005	1.081	0.988
Cow milk + Khlass (10%)	−0.0004	1.067	0.985
Camel milk + Sukkary (15%)	−0.0005	1.084	0.970
Camel milk + Khlass (10%)	−0.0005	1.072	0.975

**Table 5 foods-13-00524-t005:** Chemical composition of date fruits (flesh) for Sukkary and Khlass cultivars (Tamr stage of maturity) *.

Chemical Component	Sukkary	Khlass	Method/Ref.
Moisture	31.11 ^c^ ± 0.510	15.21 ^a^ ± 0.710	AOAC 2005-925.45
Units: (g/100 g dm) **			
Crude Protein	3.37 ^c^ ± 0.050	2.35 ^b^ ± 0.130	AOAC 2005-920.152
Total Fat	0.14 ^a^ ± 0.010	0.13 ^a^ ± 0.040	AOAC 2005-989.05
Crude Fiber	4.13 ^b^ ± 0.230	3.97 ^b^ ± 0.310	AOAC 2005-962.09
Ash	1.65 ^a^ ± 0.180	1.80 ^a^ ± 0.100	AOAC 2005-930.30
Total Carbohydrate	90.70 ^a^ ± 0.360	92.13 ^b^ ± 0.500	CALCULATION
Total Sugars	79.83 ^a^ ± 1.460	88.27 ^b^ ± 1.950	AOAC 2005-977.20
Fructose	7.52 ^a^ ± 0.160	42.36 ^b^ ± 0.160	AOAC 2005-977.20
Glucose	8.90 ^a^ ± 0.590	45.55 ^c^ ± 0.930	AOAC 2005-977.20
Sucrose	63.40 ^c^ ± 0.750	0.36 ^a^ ± 0.110	AOAC 2005-977.20
Units: (kcal/100 g)			
Total energy	377.59 ^a^ ± 1.700	379.11 ^ab^ ± 2.550	CALCULATION
Units: (mg/100 g dm)			
Calcium	67.40 ^c^ ± 6.830	52.38 ^ab^ ± 11.280	AOAC 2005-985.35
Phosphorus	90.91 ^b^ ± 13.280	54.15 ^a^ ± 1.400	AOAC 2005-985.35
Sodium	9.27 ^a^ ± 2.710	12.03 ^a^ ± 4.000	AOAC 2005-985.35
Potassium	687.21 ^a^ ± 32.690	682.58 ^a^ ± 55.850	AOAC 2005-985.35
Magnesium	77.88 ^b^ ± 3.890	60.35 ^a^ ± 1.960	AOAC 2005-985.35
Iron	1.35 ^a^ ± 0.020	1.27 ^a^ ± 0.030	AOAC 2005-985.35
Total soluble solids (Brix)	76.23 ^a^ ± 0.550	88.17 ^c^ ± 0.060	AOAC 2005-983.17
pH-10% @	7.70 ^c^ ± 0.030	6.31 ^a^ ± 0.150	AOAC 2005-981.12
Acidity as Citric Acid (mg/100 g dm)	0.27 ^a^ ± 0.090	0.34 ^a^ ± 0.040	AOAC 2005-942.15

* The same letter in a row in same group means no significant difference at *p* < 0.05. ** dm-dry matter.

**Table 6 foods-13-00524-t006:** Mean values of the chemical components of the fresh milk, date syrup, and the four milk/date syrup drinks.

Chemical Components	Fresh Milk		Date Syrup		Milk Drinks				Unit	Method/Ref.
	Cow	Camel	Sukkary	Khlass	Cow Milk + Sukkary 15%	Cow Milk + Khlass 10%	Camel Milk + Sukkary 15%	Camel Milk + Khlass 10%	Units	Reference
Total Carbohydrate	15.6 ^b^ ± 0.010	15.71 ^a^ ± 0.010	75.76 ^a^ ± 0.010	75.05 ^b^ ± 0.010	15.95 ^b^ ± 0.030	12.61 ^d^ ± 0.030	16.09 ^a^ ± 0.030	12.62 ^c^ ± 0.030	(g/100 g)	Calculation
Total Sugar	4.72 ^b^ ± 0.010	4.83 ^a^ ± 0.010	53.83 ^a^ ± 0.010	53.05 ^b^ ± 0.010	15.01 ^b^ ± 0.010	12.48 ^c^ ± 0.010	15.67 ^a^ ± 0.010	12.38 ^d^ ± 0.10	(g/100 g)	AOAC 2005-977.20
Fructose	0.000	0.000	13.08 ^a^ ± 0.010	26.54 ^b^ ± 0.010	2.6 ^d^ ± 0.010	3.95 ^a^ ± 0.010	2.66 ^c^ ± 0.010	3.65 ^b^ ± 0.010	(g/100 g)	AOAC 2005-977.20
Glucose	0.000	0.000	8.09 ^b^ ± 0.010	26.51 ^a^ ± 0.010	1.84 ^d^ ± 0.010	4.18 ^a^ ± 0.010	2.09 ^c^ ± 0.010	4.07 ^b^ ± 0.010	(g/100 g)	AOAC 2005-977.20
Sucrose	0.000	0.000	29.09 ^a^ ± 0.010	0.10 ^b^ ± 0.010	5.91 ^a^ ± 0.010	0.1 ^c^ ± 0.010	5.84 ^b^ ± 0.010	0.1 ^c^ ± 0.010	(g/100 g)	AOAC 2005-977.20
Maltose	0.000	0.000	3.57 ^a^ ± 0.010	0.10 ^b^ ± 0.010	1.26 ^b^ ± 0.010	0.45 ^d^ ± 0.010	1.41 ^a^ ± 0.010	0.54 ^c^ ± 0.010	(g/100 g)	AOAC 2005-977.20
Lactose	4.72 ^b^ ± 0.010	4.83 ^a^ ± 0.020	0.000	0.000	3.40 ^d^ ± 0.020	3.80 ^b^ ± 0.020	3.67 ^c^ ± 0.020	4.02 ^a^ ± 0.020	(g/100 g)	AOAC 2005-977.20
Proteins	3.06 ^b^ ± 0.010	3.10 ^a^ ± 0.10	1.47 ^a^ ± 0.010	0.88 ^b^ ± 0.010	2.90 ^c^ ± 0.010	2.92 ^b^ ± 0.010	2.88 ^d^ ± 0.010	2.93 ^a^ ± 0.010	(g/100 g)	FOSS-AN-300
Caseins	2.45 ^a^ ± 0.010	2.25 ^b^ ± 0.010	0.10 ^a^ ± 0.010	0.10 ^a^ ± 0.010	2.15 ^b^ ± 0.010	2.30 ^a^ ± 0.010	1.87 ^d^ ± 0.010	2.03 ^c^ ± 0.010	(g/100 g)	AOAC 2005-998.06
Crude Fiber	0.10 ^a^ ± 0.010	0.10 ^a^ ± 0.010	0.10 ^a^ ± 0.001	0.10 ^a^ ± 0.001	0.1 ^a^ ± 0.001	0.1 ^a^ ± 0.001	0.1 ^a^ ± 0.001	0.1 ^a^ ± 0.001	(g/100 g)	AOAC 2005-962.09
Fat	2.95 ^a^ ± 0.020	2.86 ^b^ ± 0.010	0.10 ^a^ ± 0.001	0.10 ^a^ ± 0.001	2.64 ^b^ ± 0.001	2.78 ^a^ ± 0.001	1.75 ^d^ ± 0.001	1.81 ^c^ ± 0.001	(g/100 g)	AOAC 2005-989.05
Ash	0.69 ^b^ ± 0.010	0.82 ^a^ ± 0.010	1.42 ^a^ ± 0.010	1.49 ^a^ ± 0.010	0.71 ^c^ ± 0.010	0.60 ^d^ ± 0.010	0.86 ^a^ ± 0.010	0.83 ^b^ ± 0.010	(g/100 g)	AOAC 2005-930.30
Calcium	938 ^b^ ± 1.000	1125 ^a^ ± 1.000	494 ^a^ ± 1.000	424 ^b^ ± 10.000	867 ^d^ ± 10.000	885 ^c^ ± 1.000	965 ^b^ ± 1.000	1002 ^a^ ± 1.000	ppm	AOAC 2005-985.35
Magnesium	99 ^a^ ± 0.000	78 ^b^ ± 2.000	427 ^a^ ± 1.000	420 ^a^ ± 1.000	142 ^a^ ± 1.000	123 ^b^ ± 1.000	123 ^b^ ± 1.000	106 ^c^ ± 1.000	ppm	AOAC 2005-985.35
Sodium	293 ^b^ ± 1.000	383 ^a^ ± 1.000	212 ^a^ ± 1.000	164 ^b^ ± 1.000	297 ^c^ ± 1.000	289 ^d^ ± 1.000	357 ^b^ ± 1.000	372 ^a^ ± 1.000	ppm	AOAC 2005-985.35
Potassium	1382 ^b^ ± 1.000	1827 ^a^ ± 0.000	0.53 ^a^ ± 1.000	0.54 ^a^ ± 1.000	1967 ^c^ ± 1.000	1702 ^d^ ± 1.000	2242 ^a^ ± 1.000	2082 ^b^ ± 1.000	ppm	AOAC 2005-985.35
Standard acidity (Lactic acid)	0.08 ^a^ ± 0.100	0.08 ^a^ ± 0.200	1 ^a^ ± 0.200	1 ^a^ ± 0.100	0.26 ^a^ ± 0.100	0.21 ^c^ ± 0.100	0.22 ^b^ ± 0.100	0.21 ^c^ ± 0.100	(g/100 g)	AOAC 2005-935.57
Vitamin A	56.8 ^b^ ± 0.200	70.30 ^a^ ± 0.200	1 ^a^ ± 0.100	1 ^a^ ± 0.300	430.4 ^b^ ± 0.200	556.50 ^a^ ± 0.200	43.17 ^c^ ± 0.003	37.13 ^d^ ± 0.002	IU/100 g	HPLC-LUNN
Vitamin D	45.4 ^a^ ± 0.200	19.44 ^b^ ± 0.300	10 ^a^ ± 0.200	10 ^a^ ± 0.200	10 ^a^ ± 0.200	10 ^a^ ± 0.200	10 ^a^ ± 0.200	10 ^a^ ± 0.200	IU/100 g	HPLC-LUNN
Total energy	58 ^a^ ± 1.000	58 ^a^ ± 2.000	309 ^a^ ± 1.000	304 ^b^ ± 2.000	99 ^a^ ± 1.500	87 ^c^ ± 1.500	91.63 ^b^ ± 1.500	78.50 ^d^ ± 1.500	kcal/100 g	Calculation

The same letter in a row means no significant differences based on the least square difference (LSD) test and a significance level at *p* < 0.05.

**Table 7 foods-13-00524-t007:** PLSR performance for M.C., aw, TSS, pH, and BI properties of the favored milk drink (FD), and TSS and BI properties for fresh milk (FM) and Khlass date syrup (DSK) for both calibration and cross-validation models.

Parameter	Calibration		Cross-Validation	
	R^2^	RMSRC	R^2^	RMSECV
M.C._FD_	0.988	0.777	0.982	0.788
a_wFD_	0.984	0.746	0.996	0.764
TSS_FD_	0.941	0.727	0.687	0.727
pH_FD_	0.957	0.755	0.955	0.723
BI_FD_	0.978	0.743	0.988	0.703
TSS_FM_	0.948	0.680	0.948	0.680
BI_FM_	0.978	0.762	0.946	0.643
TSS_DSK_	0.958	0.603	0.965	0.603
BI_DSK_	0.966	0.625	0.942	0.557

**Table 8 foods-13-00524-t008:** ANNs performance for M.C., aw, TSS, pH, and BI properties for the favorable milk drink (FD), and TSS and BI properties for fresh milk (FM) and Khlass date syrup (DSK) for both calibration and cross-validation models.

Parameter	Calibration		Cross-Validation	
	R^2^	RMSRC	R^2^	RMSECV
M.C._FD_	0.989	0.745	0.989	0.744
a_wFD_	0.984	0.755	0.984	0.725
TSS_FD_	0.946	0.715	0.946	0.754
pH_FD_	0.955	0.740	0.955	0.711
BI_FD_	0.978	0.735	0.978	0.713
TSS_FM_	0.948	0.684	0.948	0.648
BI_FM_	0.978	0.715	0.978	0.633
TSS_DSK_	0.959	0.592	0.959	0.598
BI_DSK_	0.966	0.611	0.966	0.547

## Data Availability

The data presented in this study are available on request from the corresponding author. The data are not publicly available due to privacy restrictions.

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
