# Peer review of "Non-Destructive Evaluation of the Physiochemical Properties of Milk Drink Flavored with Date Syrup Utilizing VIS-NIR Spectroscopy and ANN Analysis"

_foods, 2024, doi:10.3390/foods13040524_

Round 1
Reviewer 1 Report
Comments and Suggestions for Authors
Date syrup is a delicious sweetener which in this case is used as a blender of milk to make it sweeter and tastier. The objective of this work is to measure physiochemical properties of dates fruits and milk drinks flavored with dates syrup in different concentration (5, 10, 15, and 20 gm), and to evaluate the physical properties of milk drinks utilizing near-infrared spectroscopy. Partial least-square regression and artificial neural network analysis are used for processing the spectroscopic data.
The practical applications of this study are clearly listed at the beginning of the paper, being the most relevant the non-destructive testing in industrial settings, where both both cow and camel milks are processed.
Density at different temperatures, color, chemical analyses and sensory evaluation were obtained by standard instruments and considered as reference data for the spectroscopic data processing and model building. Spectroscopy data were collected in the 310–1100 nm band by means of a handheld device.
The correlation coefficients of prediction by comparing the reference spectroscopy data were reported and resulted very good (higher than 0.9).
The following are my comments for a minor revision:
1) as the entire VIS-NIR 310–1100 nm band is used, the title should mention that (not only NIR). Please change.
2) it is not specified the number of samples used for calibration and validation. Please add.
3) Table 4 generally mentions R2. Would it be possible to list it for calibration/validation and the related errors ??
Reviewer 2 Report
Comments and Suggestions for Authors
The manuscript “Non-destructive evaluation of the physiochemical properties of milk drink flavored with date syrup utilizing NIR Spectroscopy and ANN analysis” presents results of measurement of the physiochemical properties of dates fruits and milk drinks flavored with date syrup and evaluation of the physical properties of milk drinks utilizing a non-destructive NIR spectra. The topic is interesting, but the manuscript is poorly written. My biggest concern is the small number of prepared samples. Taking into account that only samples were prepared, ANN modeling of the sample properties makes no sense. Furthermore, there is a complete lack of discussion of the presented results and comparison with the available literature. The results and discussion section were complexly written without comparison with the available literature.
Practical application: Please explain the sentence that NIR spectrophotometry is a tool for forecasting physiochemical properties.
All Latin expressions should be written in italics, for example, Phoenix dactylifera.
Introduction section: The motivation for the research and novelty of the research are not clear.
2.2. Methods, - delete the comma.
Add the numeration of subsections in the method section.
The major unit operations include the removal of date pits, mechanical mincing to produce date paste, and mixing and heating of the date paste with water (at a w/w ratio of 2.5:1 (paste: water(weight/weight)) to produce a homogenized date suspension [40, 41]. Remove the outer bracket.
Date syrup can be preserved for several months at room temperature since the sugar content reaches 80%. Please add a reference to that statement.
Please define the number of replications for physicochemical evaluation, chemical evaluation, and sensory evaluation.
Please clarify the title, “Evaluation of physical properties of milk drinks utilizing NIR”. Why was the second derivative method used for pre-processing? Specify the data matrix dimension for PLS modeling. What were the model output variables?
Statistical analysis: How was data normability tested?
Results: What are NIR models?
All results in the manuscript should be presented with standard deviations or standard errors. Presenting statistical differences between samples without evident standard deviations is meaningless.
Table 3 and Figure 2 present the same data.
Table 4. Please improve the table wrapping.
NIR spectra: Please present the raw NIR spectra (spectra without pre-processing).
What cross-validation method was used for ANN modeling?
Whole spectra were used for modeling?
The authors described the preparation of four samples of milk, so it is not clear how they performed ANN modeling. Four data points in the output layers are clearly too small a data set for any kind of modeling!
Discussion. There is a complete lack of discussion of the presented results and comparison with available literature. The results and discussion section were complexly written without comparison with the available literature.
Reviewer 3 Report
Comments and Suggestions for Authors
This paper investigates the use of a non-destructive evaluation of the physiochemical properties of milk drink flavored with date syrup utilizing NIR Spectroscopy and ANN analysis. The methods utilized in this study are highly specific, and the experimental procedures are meticulously detailed. Moreover, the statistical analyses conducted are adequate. The experiments have been well-conceived and executed.
However, there are several areas in which the authors can enhance the manuscript:
Material and methods
“Preparation of milk drinks flavored with date syrup” d
ate syrup (5, 10, 15, and 20 gm), gm is gram? If yes use “g” and not “gm” the same in “milk (95, 90, 85, and 80 gm” and in “100 gm”. Please correct other situations throughout the manuscript
a) Physical properties
“48 hrs.” change to “48 h”
Results
Table 1. – please indicate in the table footnote was is (w.b.)
Table 1 and Table 2 – please pay attention to the number of significant figures for each of the parameters presented in the tables
Table 3 and Figure 2 – the density values for the four milk drinks at different temperatures are presented in Table 3 and Figure 2, please choose one. If necessary, you could use the supplementary material.
3.2 Sensory evaluation of all samples of milk drinks flavored with date syrup
Are there significant differences among the samples in terms of sensory evolution? If yes, please indicate them.
Table 7 and table 8. in the parameters “AW” change to “aw”
The discussion of results should also be improved, and the data obtained should be compared and discussed, whenever possible, with the information presented in the literature.
Reviewer 4 Report
Comments and Suggestions for Authors
Overall comment:
Overall, the authors prepared and evaluated a milk drink flavored with dates syrup produced at a lab-scale level. The date syrup production involves a sequence of essential unit operations, commencing with the extraction, filtration, and concentration processes from two cultivars: Sukkary and Khlass. Date syrup was then mixed with cow’s and camel’s milk at four percentages. Based on the formulation, sensory, physical, and chemical characteristics of the milk drinks flavored with date syrup were examined. Besides, the work assessed the characteristics of the milk drink enhanced with date syrup by employing near-infrared spectra (NIR) spectra and analyzed utilizing partial least-square regression (PLSR) and artificial neural network (ANN) analysis.
Comment 1: Many date flavored milk products can be found with a single google search in Saudi Arabia. Can the authors illustrate more about the innovative of the work?
Comment 2: In the introduction part, the authors should describe in details and specificity what physical properties the NIR is used to characterize.
Comment 3: Why did the authors spend so much effort characterizing the physical properties of the date flavored milk? What is the purpose of doing that? While the NIR characterization is not well presented and studied.
Comment 4: Figure 3 can be enlarged with a larger font for a better visualization.
Comments on the Quality of English Language
Minor English editing is needed.
Round 2
Reviewer 2 Report
Comments and Suggestions for Authors
The authors partially answered my comments, and there are still important issues to discuss. It is not clear how the authors performed ANN modeling and what the dimension of the data set was. My suggestion is a major revision. Specific comments are listed below.
ANN modeling: please specify the data matrix dimension used for the modeling. What were the model inputs? Whole VIS-NIR spectra? Specify the number of experimental data points in the ANN input layer. The authors mentioned in the manuscript that calibration models were then established using a dataset comprising 3600 samples. It is not clear how they got that number when only milk drinks flavored with date syrup were prepared.
All experimental results (Tables 1–6) in the manuscript should be presented with standard deviations or standard errors. Standard deviations are typically reported as descriptive statistics to provide a sense of how widely dispersed the observations are around the mean. For each table, specifically describe in the table caption how the data was presented and list the number of replications used for the mean value calculation.
Table 5. Specifically describe in the table caption how the data were presented and list the number of replications used for the mean value calculation.
Reviewer 4 Report
Comments and Suggestions for Authors
The authors addressed the issues.
Author Response
Note to Reviewer 4
- The authors addressed the issues.
- Thanks for the reviewer for the comment. Indeed, to current time, no published studies have comprehensively quantified both the objective and subjective aspects of the physicochemical properties and sensory evaluation of milk beverages enhanced with date syrup through advanced analysis employing NIR and ANN techniques. The aim of this study was to develop and assess the physiochemical properties of milk beverages flavored with different concentrations of date syrup variants (Khlass and Sukkary) using NIR spectra alongside PLSR and ANN analyses. Non-destructive NIR techniques offer the advantage of real-time, online application in the food and dairy industries, unlike conventional methods. This technology aims to simplify sensory and routine analyses for fresh produce, as well as during processing and storage, resulting in reduced time, cost, and labor in future projects.
- This paper presents a methodology for enhancing cow and camel milk with Khlass and Sukkary date cultivars using NIR technology. The study demonstrates the potential of this approach in processing and highlights opportunities for innovating and enhancing dairy products.
Thanks again to the reviewer for the review and the satisfaction of our previous answers/modifications.